# Transformers and slot encoding for sample efficient physical world modelling

## Abstract

World modelling, i.e. building a representation of the rules that govern the world so as to predict its evolution, is an essential ability for any agent interacting with the physical world. Recent applications of the Transformer architecture to the problem of world modelling from video input show notable improvements in sample efficiency. However, existing approaches tend to work only at the image level thus disregarding that the environment is composed of objects interacting with each other. In this paper, we propose an architecture combining Transformers for world modelling with the slot-attention paradigm, an approach for learning representations of objects appearing in a scene. We describe the resulting neural architecture and report experimental results showing an improvement over the existing solutions in terms of sample efficiency and a reduction of the variation of the performance over the training examples. The code for our architecture and experiments is available at **[Redacted from the anonymized version]**

## 1 Introduction

World modelling is the ability of an artificial agent to build an internal representation of the world in which it operates. This representation is employed by the agent to forecast the evolution of the world. The problem of building a world model spans many branches of artificial intelligence, such as planning and reinforcement learning (Micheli et al., 2023; Paster et al., 2021), physics modelling and reasoning (Ding et al., 2021), and robotics (Wu et al., 2022). An accurate representation of the world allows building simulations that, in turn, enable practitioners to gather additional data and test the performance of an artificial agent without interacting with their environment. This is convenient because interacting with an agent's environment can be time-consuming, risky due to possible failures of physical components, and sometimes even impossible due to the potential unavailability of the environment (e.g. experimenting with the exploration of Mars by a rover).

Recent applications of the Transformer architecture (Vaswani et al., 2017) to the task of world modelling from video input suggest that this family of architectures is not only it is capable of capturing the dynamics of the environment, but it is also capable of learning with high sample efficiency (Micheli et al., 2023; Robine et al., 2023). However, existing approaches typically operate directly at the image level, with little regard for the objects contained within it. Understanding how objects interact with each other and with the environment is of paramount importance, as it endows agents with an intuitive theory of object motion (McCloskey, 1983). In neural architecture, this problem is addressed in a separate line of research (Locatello et al., 2020; Kipf et al., 2022; Wu et al., 2023), that focuses on learning object-based representations that allow the objects in the scene and their interactions to be modelled explicitly. We hypothesise that Transformers may benefit from object-based representations to learn more accurate models of the world.

**Problem statement** In this paper, we focus on physical world modelling through the analysis and prediction of synthetic videos. Learning a model of basic physical laws such as gravity and collision is very important for any agent working in a real environment to understand better the world's evolution and the consequences of the agent's actions.

Machine learning research on modelling intuitive physics aims to replicate the innate understanding of physical concepts that humans display since their first months of age (McCloskey et al., 1983; Baillargeon, 2004). Specifically, we approach the physical interaction output prediction problem, as

defined in a recent survey (Duan et al., 2022). In this context, the agent is shown a video, composed of a sequence of frames $x_1, ..., x_T$ depicting several objects interacting in a world governed by physical laws such as gravity and collision. The agent is then asked to predict the final outcome of the situation, which requires estimating how the objects will behave after what is shown in the input video (Yi et al., 2020).

**Contribution** We design a transformer-based architecture for world modelling, inspired by the principles of representation learning with slot encoding (Kipf et al., 2022; Singh et al., 2022). The evaluation is based on how well the learned model predicts the outcome of the situations it gets shown. We show that this allows us to reap the benefits from both approaches (i.e. slot encoding and transformers), while also noticeably improving the stability of the training process.

**Paper outline** We structure the rest of the paper as follows. Section 2 discusses some previous works on the topics of world modelling and representation learning. Section 3 describes our architecture and how it was trained. Section 4 describes the evaluation experiments we performed. Section 5 offers some final remarks on this work.

## 2 RELATED WORK

### 2.1 WORLD MODELLING

The world modelling problem has received tremendous attention in the last few years. In reinforcement learning, being able to simulate the environment dynamics is especially useful, because it enables the agent to act and learn in its own simulated world without paying the cost of interacting with the "real" environment. The Dreamer algorithm, in its various versions (Hafner et al., 2020; 2021; 2023), has been relatively influential on the topic, with Wu et al. (2022) being an application in a robotic domain, where real-world interaction has the unfortunate potential of breaking usually costly equipment, in addition to time costs. Additional approaches in reinforcement learning include solutions based on causal discovery and reasoning (Yu et al., 2023), which aim at learning a causal model of the environment to better understand the interactions between the agent and the world, and even provide an explanation for the actions taken by the agent. Transformer-based approaches are also studied, due to their generally good performance in different tasks and the sample efficiency they provide in this specific problem (Micheli et al., 2023; Robine et al., 2023).

In the case of agents acting in a real physical environment, learning a model of the basic physical laws of the world is essential to act effectively in the environment and understand the consequences of each move. We refer to this problem as intuitive physics modelling. For this reason, several solutions have been studied for this problem with approaches ranging from deep learning (Qi et al., 2021), to violation of expectation (Piloto et al., 2022), to causal reasoning (Li et al., 2022).

With this work, we aim to improve the general level of performance and stability of Transformer-based approaches by implementing an unsupervised representation learning module, specifically one based on the principles of slot encoding.

### 2.2 OBJECT-ORIENTED REPRESENTATION LEARNING

In this work, we integrate concepts from the line of research on learning object-oriented representations from images and videos (Locatello et al., 2020; Zoran et al., 2021; Jia et al., 2023). In particular, slot encoders for video (Kipf et al., 2022; Singh et al., 2022) learn a representation that tracks the prominent objects in a video frame by frame. The structure of our architecture is based on that of slot encoders, but we try to streamline it by focusing exclusively on using Transformer modules, which allows us to convert the image to just one intermediate form, i.e. a sequence of tokens to be elaborated by the Transformers, while Singh et al. (2022) requires two separate elaborations of the image: one to produce convolutional features and one to produce a sequence of tokens.

The idea of leveraging slot encoding mechanisms for world modelling has also been explored in Wu et al. (2023). However, while that work uses a single transformer as the dynamics modelling module, we experiment with the idea of keeping representation correction and dynamics advancement separate, where each step is learned by a different, smaller neural model.

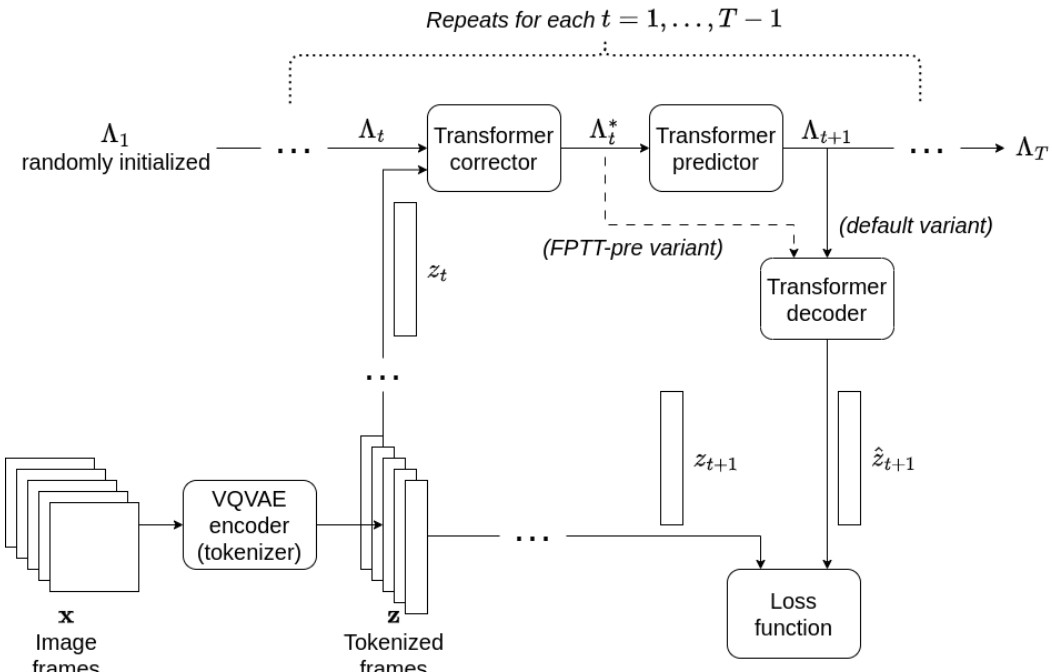

Figure 1: Architecture diagram for the Future-Predicting Transformer Triplet for world modeling.

This is not the only existing approach: an additional, earlier line of work focuses on using generative models with object-centric features for images (Eslami et al., 2016; Engelcke et al., 2020) and video (Kosiorek et al., 2018) to distinguish the objects present in a scene and improve image/video generation with the learned object awareness. Jiang et al. (2020) applies this paradigm to world modelling. Other approaches include spatial attention (Lin et al., 2020) and latent space factorization (Kabra et al., 2021).

## 3 METHOD

We introduce a multi-stage architecture, called "Future-Predicting Transformer Triplet for world modeling" (**FPTT**), which aims to model the behaviour of objects in a set of videos so as to predict their evolution.

We frame the world modelling problem as a sequence learning one. We use transformers as the fundamental building block of our architecture, due to their proven performance in world modelling (Micheli et al., 2023) as well as other tasks that can be reduced to modelling and manipulating a sequence of tokens (Vaswani et al., 2017; Esser et al., 2021). In particular, we leverage the recent work by Micheli et al. (2023), which showcases a sample-efficient application of transformers to world modelling. This design principle is integrated with the slot-attention mechanism (Kipf et al., 2022; Singh et al., 2022), which is used to learn a compact representation of objects that appear in a video.

### 3.1 NOTATION

We indicate with $x_t$ the $t$-th frame of a video and with $z_t$ a sequence of tokens corresponding to $x_t$. $\Lambda_t(x)$ is the internal representation of the input video $\mathbf{x}$ up to time $t$, i.e. given frames $x_1, ..., x_{t-1}$, while $\Lambda_t^*(x)$ is the "corrected" representation which also includes information from frame $x_t$. The initial representation $\Lambda_1(x)$ is randomly determined for initialization purposes.

## 3.2 ARCHITECTURAL OVERVIEW

Figure 1 shows the high-level components of FPTT. Further details on the implementation, e.g. the hyperparameters of the architecture, can be found in Appendix A.

The architecture takes as input a sequence of $T$ frames of a video, i.e. $x_t$ with $t = 1, ..., T$. The frames are processed sequentially, so that $\Lambda_{t+1}(x)$ is determined by combining the previous representation $\Lambda_t(x)$ with the new frame $x_t$.

As in previous works (Micheli et al., 2023), each frame $x_t$ is transformed into a corresponding sequence of tokens $z_t$ by a discrete Vector Quantized Variational Autoencoder (VQVAE) (van den Oord et al., 2017; Esser et al., 2021), as transformers need to work on sequences of tokens. Further details on the VQVAE can be found in section 3.3.

After this preliminary step, the sequence of tokens $z_t$ is processed by the core components of the architecture which are meant to predict the next representation $\Lambda_{t+1}(x)$ based on the current one $\Lambda_t(x)$ and $z_t$. These components, both based on the transformer architecture, have the same high-level purpose as their counterparts in slot attention for video (Kipf et al., 2022; Singh et al., 2022) architectures:

- The **corrector transformer** (see section 3.4) which compares the previous (internal) representation $\Lambda_t(x)$ with the tokenized representation of the current frame $z_t$ in order to consistently align the internal representation with the actual evolution of the video;

- The **predictor transformer** (see section 3.5) which predicts the evolution of the world state and produces the representation of the next time step $\Lambda_{t+1}(x)$ on the basis of the result of the corrector, i.e., $\Lambda_t^*(x)$. $\Lambda_{t+1}(x)$ is then passed to the corrector for the next stage.

The result of the prediction at stage $t$, i.e. $\Lambda_{t+1}(x)$, is also passed to the **decoder transformer** (see section 3.6). This component transforms the predicted internal representation $\Lambda_{t+1}(x)$ into a sequence of tokens $\hat{z}_{t+1}$. Finally, the loss is calculated by comparing $\hat{z}_{t+1}$ with $z_{t+1}$, i.e. the sequence of tokens obtained from the input frame. All the above steps (correction, prediction, decoding and loss calculation) are computed for each input frame except the last one, i.e. for $t = 1, ..., T - 1$. The last frame is not processed at training time because it would require the existence of a frame $x_{T+1}$ to calculate the loss against, which is impossible to provide since the video only goes up to frame $x_T$ by definition.

## 3.3 VECTOR QUANTIZED VARIATIONAL AUTOENCODER FOR TOKENIZATION

The Vector Quantized Variational Autoencoder (VQVAE) transforms video frames into a format that subsequent transformers can process. This format is a sequence of $L$ tokens, with each token represented by a vector of the space $\mathcal{V} = \{v_1, v_2, ..., v_N\} \subset \mathbb{R}^d$, where $d$ is defined as a hyperparameter of the architecture (see appendix A).

The VQVAE alternates residual convolutional layers, attention blocks, and convolutional downsampling layers to convert an image[1] $x \in \mathbb{R}^{W \times H \times 3}$ (W and H are the width and height of the image, respectively) to a latent-space representation $z_l(x) = (z_{l,1}(x), z_{l,2}(x), ..., z_{l,L}(x)) \in \mathbb{R}^{L \times d}$. Then each latent vector is quantized into a token simply by picking the closest embedding vector in $\mathcal{V}$, that is to say, $z(x) \in \mathbb{R}^{L \times d}$ is such that $z_i(x) = argmin_{v \in \mathcal{V}} (||z_{l,i}(x) - v||_2)$ for each $i = 1, ..., L$.

A decoder network with a symmetrical structure to the encoder (not shown in figure 1) is used to reconvert a token sequence $z$ back into an image $\hat{x}(z)$ for the purposes of training the whole autoencoder pair.

## 3.4 CORRECTOR TRANSFORMER

The purpose of the corrector transformer is to avoid drifting, i.e. making the internal representation stick with the evolution of the video. This is achieved by updating the estimated representation $\Lambda_t(x)$ with the corresponding frame $z_t$ thus producing a corrected representation $\Lambda_t^*(x)$. It is implemented

---

[1]We omit the $t$ subscript in this paragraph for ease of notation, since the VQVAE processes images as single entities, not as parts of a video.

by a transformer that produces the corrected representation ($\Lambda_t^*(x)$) by performing an unmasked cross-attention of the two inputs ($\Lambda_t(x)$ and $z_t$).

It is worth noting that this structure fits neither the transformer encoder nor the transformer decoder descriptions as traditionally defined in Vaswani et al. (2017), since we perform cross-attention (like a decoder) without including a causal mask (like an encoder). This allows us to compare the two input sequences as a whole, without arbitrarily limiting the context. We also note that the purpose of this transformer is not to perform autoregressive generation, so a non-causal flow of information causes no harm.

## 3.5 PREDICTOR TRANSFORMER

The predictor transformer performs self-attention on the representation $\Lambda_t^*(x)$ to estimate its advancement to the next time step $\Lambda_{t+1}(x)$.

The predictor and corrector transformers can be seen as two halves of one model, dedicated to predicting the next internal representation on the basis of the current representation and the current frame of the video being processed. For this reason, each of them has individually fewer layers compared to the decoder (see section 3.6).

## 3.6 DECODER TRANSFORMER

The decoder transformer converts a representation $\Lambda_{t+1}(x)$ into a sequence of tokens $\hat{z}_{t+1}$. The loss is computed by comparing $\hat{z}_{t+1}$ with $z_{t+1}$, i.e. the sequence of tokens obtained from the input frame.

## 3.7 ARCHITECTURE VARIANTS

We experiment with two variants of this architecture, defined by the positioning of the decoder transformer in the structure laid out so far.

The default **FPTT** architecture, represented by the continuous lines in figure 1, has this stage positioned between the predict step that generates $\Lambda_{t+1}(x)$ and the subsequent correct step. Thus, the loss computes the error on the prediction of the frame $z_{t+1}$.

The alternative variant, which we call **FPTT-pre**, is identical except that the decoder transformer takes $\Lambda_t^*(x)$ instead. This is represented in figure 1 by replacing the line labeled as "default" with the dashed line labeled as "FPTT-pre". This produces a structure that is more in line with Kipf et al. (2022) and Singh et al. (2022), while the default FPTT diverges from such previous work.

The objective of this experiment is to test whether calculating the loss on the predicted future representation, as opposed to the corrected current one, directs the model's attention towards the accurate prediction of future events, thus emphasising the world modelling objective. This will be achieved by experimenting with different placements of the decoder transformer (and thus of the loss function).

In the absence of any contrary indication, the two variants are to be considered as operating in a similar manner.

## 3.8 TRAINING

The VQVAE is trained in isolation with respect to the whole architecture. To enhance the stability of the training process, we maintain a fixed configuration of the VQVAE parameters throughout the training of the other components. Following Micheli et al. (2023), the loss is a combination of a mean absolute error and a perceptual loss (Johnson et al., 2016) on the reconstruction, as well as a commitment loss on the embeddings (van den Oord et al., 2017).

As for the corrector, predictor and decoder transformers, they are trained together in an end-to-end fashion, with the objective to minimize a cross-entropy loss on the (tokenized versions of the) predicted frames $\hat{z}_2, ..., \hat{z}_T$ with respect to the ground-truth ones $z_2, ..., z_T$.

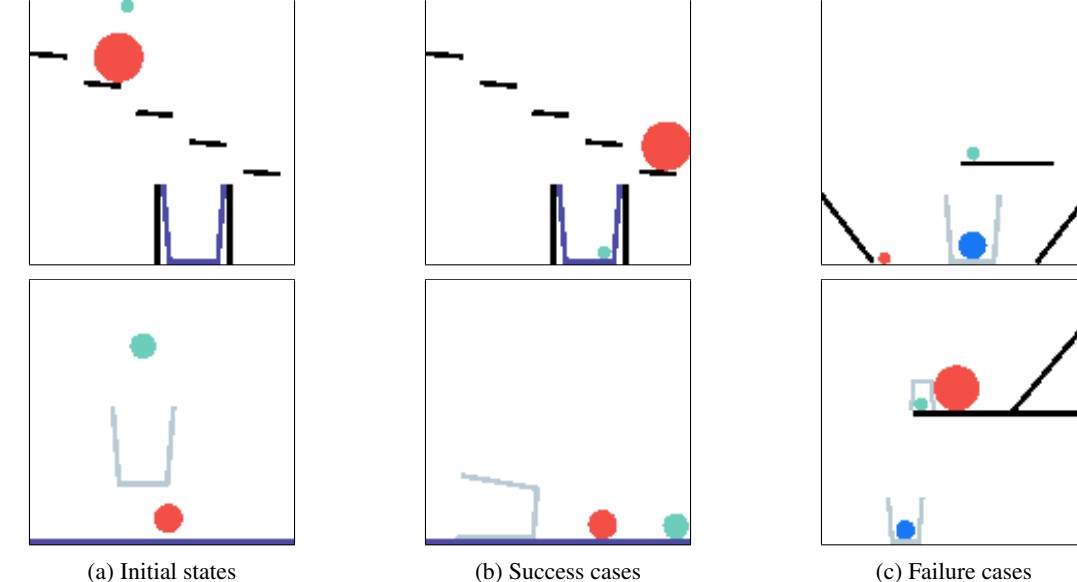

|                    |                   |                   |
| :----------------: | :---------------: | :---------------: |
| (a) Initial states | (b) Success cases | (c) Failure cases |

Figure 2: Example frames from the PHYRE dataset. 4 task types out of the full 23 are exemplified.

Both parts of the architecture are trained in a self-supervised way, with unlabeled videos from a suitable dataset. See section 4.4 for further details on the dataset.

## 4 EXPERIMENTS

The ability to model the world of the proposed architecture (FPTT) is assessed through a physical reasoning task (see section 4.1) that requires the ability to predict how a set of objects moves in a given environment. Specifically, we experiment with the PHYRE dataset which provides a benchmark containing a wide set of simple classical mechanics puzzles in a 2D physical environment (Bakhtin et al., 2019). We compare the performance of the two variants of the FPTT architecture against each other and a baseline taken from the existing literature. We also performed an ablation study to investigate the efficacy of the various components of the architecture.

### 4.1 PHYSICAL REASONING TASK

We adhere to the definition of a physical reasoning task as outlined in the PHYRE benchmark (Bakhtin et al., 2019). The task is set in a two-dimensional world that simulates simple deterministic Newtonian physics with a constant downward gravitational force and a small amount of friction. This world contains non-deformable objects, distinguished by colour, that can be static (i.e. they remain in a fixed position) or dynamic (i.e. they move if they collide with another object and are influenced by the force of gravity). These objects can be arranged in different configurations to create a wide diversity of tasks. We use a dataset made of video recordings from 23 different tasks, for a total of 1.15M video samples.

A task consists of an initial world state and a goal (see figure 2). The initial world state is a predefined configuration of objects. The goal for all tasks is the following: at the end of the simulation the green object must touch the blue object. If the goal is achieved, the task succeeds (as in the PHYRE terminology).

Given a video (as a sequence of frames), the objective of the world model is to build an internal representation that can be used to predict if the depicted task will succeed or fail. The ability to predict that represents an auxiliary classification problem. This allows us to indirectly assess the performance of the world models. It is worth noticing that the same evaluation protocol is used in related work (Wu et al., 2023).

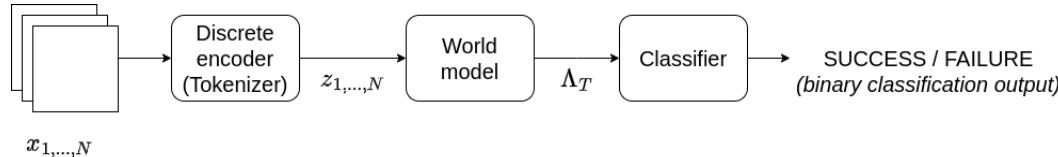

$x_{1,...,N}$

Figure 3: Diagram of the experimental setup, showing how the classifier is positioned with respect to the world modelling architecture. Note: in the case of the decoder-only ablation, replace $\Lambda_T$ with $z_T$.

## 4.2 BASELINE

The performance of the proposed architecture is evaluated in comparison to **STEVE** (Singh et al., 2022), a slot encoding architecture that is also based on the correction-prediction pattern. The input frame is encoded using a convolutional neural network (CNN) which feeds a recurrent neural network (RNN) acting as corrector. The result is then passed to the predictor (there called "interaction step"), a single-layer transformer.

For the purposes of loss calculation, the slots resulting from the corrector (before the predictor) are translated into a token sequence (i.e. $\hat{z}_t$) by a transformer decoder and compared against a ground-truth token sequence produced directly from the real video frames via a pretrained VAE.

## 4.3 ABLATION STUDY

As a further comparison, we consider an ablated version of the FPTT architecture where the corrector and predictor transformer (therefore, the components enabling slot-encoding mechanism) are removed from the architecture. This leaves only the decoder transformer to learn the entirety of the world modeling task, predicting the (tokenized version of the) next frame $z_{t+1}$ directly from the previous one $z_t$, without using an internal representation. In the following, we refer to this ablated architecture as **decoder-only**.

It can also be noted that this architecture represents a close adaptation to our non-interactive context of the approach to world modelling from visual data by Micheli et al. (2023), which also uses a single transformer to predict the future world state in a reinforcement learning setting.

This is the only possible ablation, because the loss is computed against a tokenized representation of the video frames, so the decoder is needed to produce the tokenized versions of the predicted frames.

## 4.4 EXPERIMENTAL SETUP

We experiment on a dataset of synthetic videos presented in Qi et al. (2021). This dataset was generated by rendering simulations from the PHYRE benchmark for physical reasoning (Bakhtin et al., 2019). Specifically, we focus exclusively on videos from B-tier tasks, and within-template evaluation. Figure 3 shows the experimental setup. The world model takes a video (as a sequence of frames) from the PHYRE task and builds a representation. This representation is then passed to a classifier which predicts the result of the task (i.e. success or failure). As for the classifier, we use a BERT-like encoder architecture (Devlin et al., 2019), trained in a supervised manner.

As for FPTT (default and FPTT-pre variant) and STEVE, we proceed as follows. Each video in the dataset represents a task that is labeled as either "success" or "failure". The world model is given the first $N$ frames of a video whose total length is $T$ frames, with $N < T$. The remaining $T - N$ frames are kept hidden from the model. In order to obtain the representation of the whole video, including both the given section and an estimation for the following hidden one, the $N$ given frames are processed as usual, updating the representation in the correct step and advancing it to the next timestep in the predict step. Afterwards, the remaining $T - N$ steps are projected by simply repeating the predict step, skipping the correction for the hidden frames (see figure 4). In the experiments, $N$ is set to 5, while $T$ varies depending on each video, ranging from 7 to 18, with many videos being 12-15 frames long.

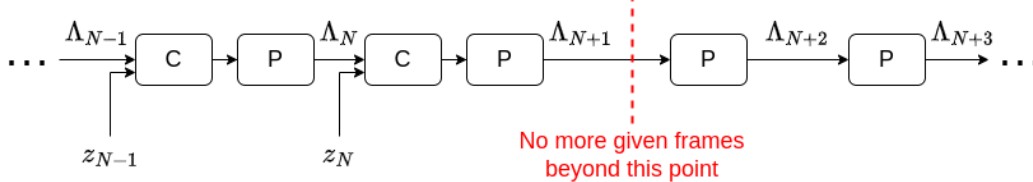

Figure 4: Illustration of the process described in section 4.4 for FPTT and STEVE. Notation has been simplified with respect to figure 1. C represents the corrector transformer, P stands for the predictor one.

We follow a similar approach for the decoder-only architecture, accounting for the lack of an internal representation in this case. The world model sees the first N frames and the remaining are generated autoregressively by the transformer. The final frame, i.e. the sequence of tokens $z_T$, is passed to the classifier instead of a representation.

All the experimenting architectures employ the same pre-trained VQVAE for transforming each input frame into a sequence of tokens (see section 3.3).

Overall, the dataset contains 1.15M videos: 95% are used for training purposes, and the remaining 5% for evaluation. We consider the following classification metrics: accuracy, precision, recall, and F1 score. The experiments were run on a server with an NVIDIA A100 graphics card, with 40GB memory.

We report the training time for experiments with the various architectures in table 1, noting that repeated runs with the same architecture resulted in extremely similar times. We attribute the much higher time in the decoder-only case to the fact that, without an internal representation, the transformer needs to deal with the longer $z_t$ sequences, and the time and memory requirements for the self-attention operation scale quadratically with sequence length.

### 4.5 RESULTS

We report on the result of our experiments in figure 5. To make reading easier, we also include versions of those plots limited to a more relevant range in figure 6. Each experiment was repeated 5 times for statistical significance.

Looking just at the plots in the figure, it can be observed that FPTT and the decoder-only the FPTT-pre variants exhibited comparable performance, while the STEVE baseline performs much worse. In all cases, the evolution of the metric value over the course of the training process is very unstable, due to the wide variety of situations proposed in the dataset. Looking closer to the F1 score (Figure 6), we can also observe that the negative peaks of FPTT are not as intense as those of the other variants, indicating a comparatively better stability.

We also claim that FPTT is more sample efficient than the baselines, and provide quantitative evidence based on Gu et al. (2017). We set a performance threshold at 0.85 on the F1 score and measure the number of training steps required in each experimental run to reach this threshold for the first time. As a consistency condition, we require the threshold to be exceeded for 6 consecutive training epochs (each epoch has 500 training steps, followed by an evaluation phase.) We report the result in table 1, noting an improvement for our default architecture with respect to the others.[2]

As for the lower performance of the STEVE baseline, we noted that during our experiments STEVE would always default to predicting a blank scene immediately after it stops being given frames (refer to the experimental setup in section 4.4), causing the class prediction to be either always "success" or always "failure". Therefore, we believe we hit a limitation of the STEVE architecture, which was conceived for slot encoding has difficulty with extrapolating a world model.

---

[2]STEVE not given a value in the S.E. column because its F1 score is significantly lower and never passes the threshold.

Table 1: Quantitative data from our experiments. "S.E." stands for "sample efficiency".

|  | S.E. Metric | Training time |
|---|---|---|
|  | *Steps (mean ± std. err.)* | *Hours* |
| FPTT (Ours) | **16200** ± 4760.8 | ∼ **3.5** |
| STEVE | N/A | ∼ **3.5** |
| Decoder only | 19000 ± 4585.3 | ∼ 8.5 |
| FPTT-pre (Ours) | 19000 ± 4937.1 | ∼ **3.5** |

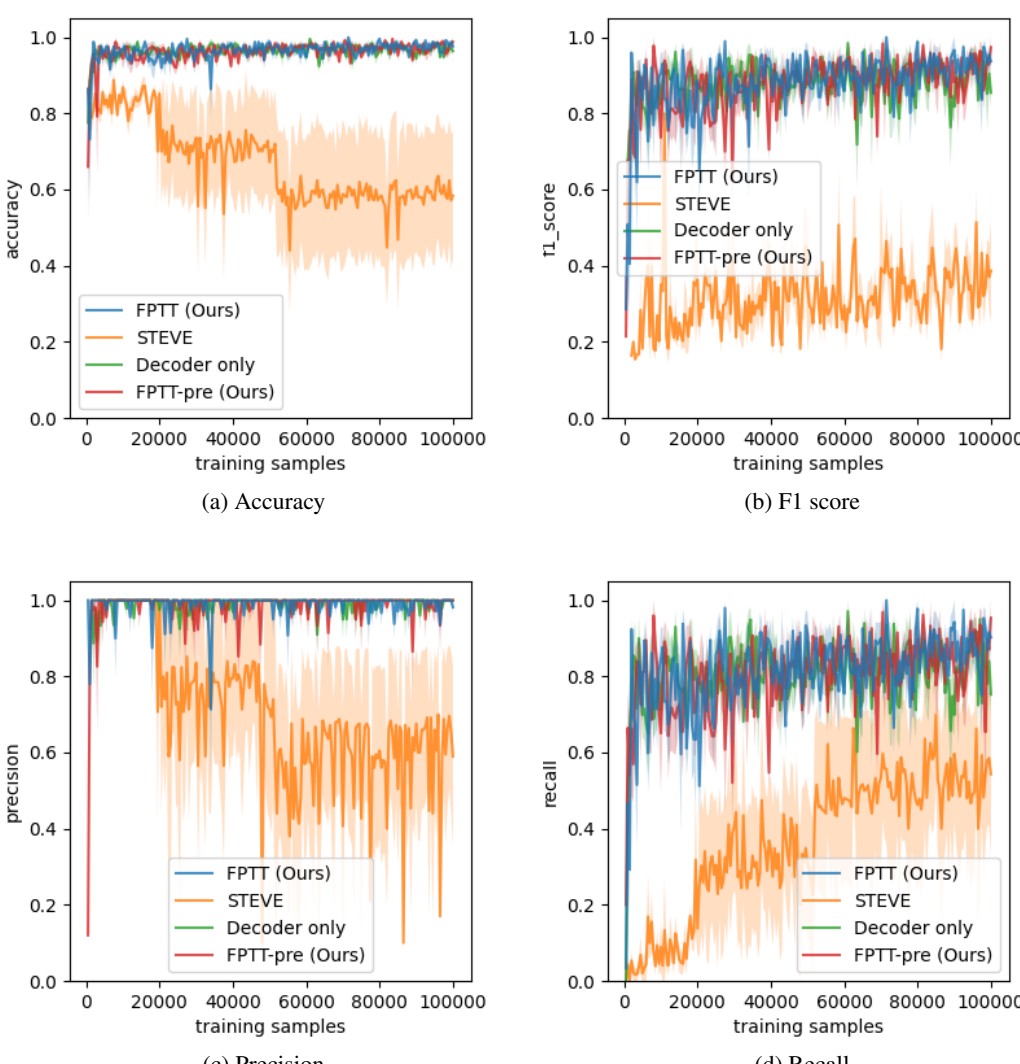

(a) Accuracy

(b) F1 score

(c) Precision

(d) Recall

Figure 5: Classification results on test data as a function of the number of training samples observed. Each line represents an average over 5 experiments; the coloured bands indicate the standard error of the mean.

## 5 DISCUSSION

In this section, we discuss the limitations of our approach, outline possible future research directions and draw our conclusions from this study.

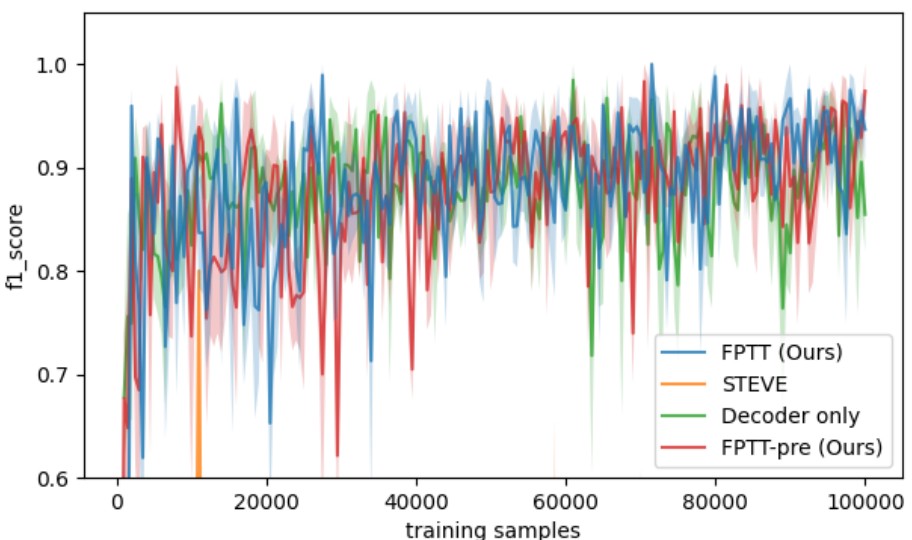

Figure 6: Classification results focused on the range [0.6, 1] on the y axis.

## 5.1 LIMITATIONS

Despite the observed performance improvements, the representation remains opaque and lacks interpretability. Our attempts at replicating the object segmentation displayed by slot-attention architectures (Kipf et al., 2022; Singh et al., 2022) have not yet yielded positive results so far. However, this may be overturned by more systematic experimentation in the future.

Furthermore, we acknowledge that, although the dataset we used presents a variety of visual configurations, it is ultimately synthetic and simplistic. Although we claim that the presented experiments demonstrate the benefit of the proposed architecture, we do plan to extend the experiments to more complex video datasets such as MOVi-E (Greff et al., 2022) and Physion (Bear et al., 2021), which will test its performance in more realistic scenes as well as its generalization capabilities. Experiments on the latter dataset are currently ongoing but could not be completed for this publication.

## 5.2 CONCLUSION

We propose a new architecture, the Future-Predicting Transformer Triplet for world modeling (FPTT), which leverages the power of transformers for sequence learning to model the behaviour of objects in a set of videos and predict the evolution of the environment.

We experimentally show that our architecture outperforms transformer-based world models (Micheli et al., 2023), and improves on slot-attention methods (Kipf et al., 2022; Singh et al., 2022) in terms of sample efficiency and stability during the training process.

In the future, we intend to conduct further experiments with the architecture in more interactive environments, in which objects can be moved by agents. Moreover, we would like to study applications to causal discovery problems (Yu et al., 2023), where learning a compact representation that can be interpreted causally might help in understanding complex scenarios.

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

## A HYPERPARAMETERS AND CONFIGURATION

This whole work was implemented in Pytorch (Paszke et al., 2019); the code is open source and published on **[Link redacted from the anonymized version, see supplementary material]**.

### A.1 VQVAE FOR TOKENIZATION

See table 2 below. The implementation is based on Esser et al. (2021) and Micheli et al. (2023).

Table 2: Hyperparameters for the VQVAE, both encoder and decoder

| Hyperparameter | Value |
|---|---|
| Video resolution (pixels) | $64 \times 64$ |
| Number of tokens per frame | 64 |
| Channels in convolution | 64 |
| Number of residual conv. layers | 10 |
| Number of self-attention layers | 3 |

### A.2 TRANSFORMERS AND SLOT ENCODING

See table 3 below. The implementation is based on nanoGPT (Karpathy, 2023). The transformers involved, i.e. the corrector-predictor-decoder triplet and the task success classifier, use the same values for the hyperparameters unless otherwise specified.

Table 3: Hyperparameters for the transformer triplet

| Hyperparameter | | Value |
|---|---|---|
| Vocabulary size | | 50304 |
| Number of tokens per frame | | 64, as above |
| Token embedding dimension | | 768 |
| Number of layers | (corrector) | 2 |
| | (predictor) | 2 |
| | (decoder) | 6 |
| | (task classifier) | 2 |
| Number of attention heads | | 12 |
| Number of slots | | 4 |
| Given video frames $(N)$ | (task classifier) | 5 |

### A.3 TRAINING PROCESS

See table 4, 5, and 6.

Table 4: General configuration for the training process

| Configuration | | Value |
|---|---|---|
| Epochs | | 100 |
| Batch size $(BS)$ | | 10 |
| Batches per epoch $(BPE)$ | | 50 |
| Training steps per epoch | | $BS \times BPE = 500$ |
| Data samples | Training | 1092500 |
| | Evaluation | 57500 |

## B EXISTING ASSET ATTRIBUTION

The following implementations have been referenced during this work:

Table 5: Optimizer for the VQVAE

| Hyperparameter | Value | |
|---|---|---|
| Type | Adam | (Kingma & Ba, 2015) |
| Leaning rate | $10^{-4}$ | |

Table 6: Optimizer for each transformer

| Hyperparameter | Value | |
|---|---|---|
| Type | AdamW | (Loshchilov & Hutter, 2019) |
| Leaning rate | $6 \cdot 10^{-4}$ | |
| Weight decay | 0.1 | |
| $(\beta_1, \beta_2)$ | $(0.9, 0.95)$ | |

- nanoGPT (Karpathy, 2023) for the Transformer implementation, licensed under the MIT license;

- Esser et al. (2021) for the VQVAE implementation, released under the MIT license;

- Micheli et al. (2023) for further details about the Transformer and VQVAE implementations, as well as the "decoder only" baseline, licensed under the GPL;

- Singh et al. (2022) for the STEVE baseline, licensed under the MIT license.

The PHYRE video dataset has been generated by Qi et al. (2021) from the PHYRE simulator (Bakhtin et al., 2019). It was downloaded following the instructions on the author's GitHub repository: `https://github.com/HaozhiQi/RPIN/blob/master/docs/PHYRE.md#11-download-our-dataset`
Correspondence with the author has confirmed that the dataset is released under the same license as PHYRE itself, i.e. the Apache license.

