# OpenReview forum: "Transformers and slot encoding for sample efficient physical world modelling"
_ICLR.cc/2025/Conference — Submitted to ICLR 2025_

### Official Review · Reviewer_AfcX · 2024-10-18

**Soundness:** 2
**Presentation:** 1
**Contribution:** 1
**Rating:** 3
**Confidence:** 5

**Summary:**

This paper proposes a world-modeling architecture that captures object-level interactions in the scene, instead of the scene itself. The architecture consists of three transformer-based models: a corrector, a predictor, and a decoder, alongside a VQ-VAE tokenizer for image encoding.

To evaluate the proposed model’s performance as a world model, the authors provide a physical reasoning task using the PHYRE benchmark, demonstrating that their model outperforms STEVE, the baseline, in terms of prediction accuracy and sample efficiency.

**Strengths:**

- The authors' approach of capturing slot-like internal representations from VQ tokens, instead of CNN embeddings, was intriguing and showed promising results.
- They propose a novel evaluation protocol for testing world model architecture, utilizing the shared benchmark with different protocols.

**Weaknesses:**

- **Lack of Novelty and Justification**: The main idea and direction of the paper have been explored in several existing works (OCVT[1], SlotFormer[2]). Although the authors are likely aware of this, they fail to convincingly demonstrate why their approach is unique and necessary for the proposed direction, compared to previous works.
- **Architecture and Design Choices**: The proposed architecture appears to be a combination of SAVi[3] and STEVE[4] architectures, but with a predictive loss function instead of a reconstructive loss function. While this variant may be promising, the paper lacks sufficient details to justify the design choices, such as thorough ablation studies. Furthermore, it is unclear whether the proposed model can outperform existing works, as it is not comprehensively compared.
- **Lack of Clarity in Architecture and Experiment Description**: The architecture section of the paper lacks clarity and detail, particularly in the description of the core components: corrector transformer, predictor transformer, and decoder transformer. Although the author provides a high-level overview of these architectural concepts, the explanation is insufficient given the emphasis on this part as the paper's core contribution. To thoroughly understand and investigate the proposed architecture, a detailed formulation of these components is necessary, including their implementation details and mathematical representations.

    Furthermore, the experiment section lacks sufficient details about the metrics used in the evaluation. To ensure transparency and reproducibility, it is essential to provide a clear explanation of each metric, including how the metric is calculated and what it represents.

- **Limited Evaluation of Proposed Architecture:** The author only provides a single task to evaluate the proposed architecture, which is insufficient to demonstrate its generality and versatility. To thoroughly assess the world modeling ability of the proposed architecture, it is essential to evaluate it on a diverse range of tasks that require the model to infer and understand the relationships between objects and scenes. Additionally, to facilitate a fair comparison with existing works, the authors may consider including several generation tasks (e.g. OBJ3D[1], CLEVR[5], Physion[6]), as has been done in prior research.

    To further demonstrate the effectiveness of the proposed model, the author could compare it with a broader range of baselines, such as SlotFormer, OCVT, SAVi, and other relevant models mentioned in the paper. Although these models are typically used for generation tasks, their predicted representations can be evaluated using the same protocol as the proposed model. Additionally, comparing with image-based world models would illustrate the advantages of object-level world models over their image-based counterparts. This approach would provide a more comprehensive understanding of the proposed model's performance and allow for a more accurate assessment of its strengths and limitations relative to existing approaches.

- **Ablation Results Raise Questions about Proposed Model:** The ablation results indicate that the ‘decoder-only’ model performs comparably to the proposed models. This suggests that VQ-tokenization and predictive loss might be sufficient to drive performance without explicitly enforcing object-level representations. This outcome seems misaligned with the paper's main theme, which emphasizes the importance of object-level representations. Consequently, this misalignment raises questions about the necessity and effectiveness of the proposed model's architecture.

[1] Wu, Yi-Fu, Jaesik Yoon, and Sungjin Ahn. "Generative video transformer: Can objects be the words?." International Conference on Machine Learning. PMLR, 2021.

[2] Wu, Ziyi, et al. "Slotformer: Unsupervised visual dynamics simulation with object-centric models." arXiv preprint arXiv:2210.05861 (2022).

[3] Kipf, Thomas, et al. "Conditional object-centric learning from video." arXiv preprint arXiv:2111.12594 (2021).

[4] Singh, Gautam, Yi-Fu Wu, and Sungjin Ahn. "Simple unsupervised object-centric learning for complex and naturalistic videos." Advances in Neural Information Processing Systems 35 (2022): 18181-18196.

[5] Johnson, Justin, et al. "Clevr: A diagnostic dataset for compositional language and elementary visual reasoning." Proceedings of the IEEE conference on computer vision and pattern recognition. 2017.

[6] Bear, Daniel M., et al. "Physion: Evaluating physical prediction from vision in humans and machines." arXiv preprint arXiv:2106.08261 (2021).

**Questions:**

- It appears that slot attention (or inverted attention) is absent, with only cross attention being mentioned. Is this an oversight in the explanation, or is it indeed absent? If it's truly absent, how can we be confident that it captures object-level dynamic understanding?

---

### Official Review · Reviewer_XAqU · 2024-11-02

**Soundness:** 3
**Presentation:** 3
**Contribution:** 2
**Rating:** 3
**Confidence:** 4

**Summary:**

The paper addresses the challenge of creating sample-efficient models for physical world modeling, focusing on predicting object interactions in dynamic environments.

The authors propose an architecture that combines Transformers with slot encoding to improve sample efficiency and stability in world modeling. Unlike existing models that operate at the image level, this model incorporates object-based representations, enabling it to capture and predict interactions more accurately.

Their model, named Future-Predicting Transformer Triplet (FPTT), uses a corrector-predictor-decoder triplet of Transformers. The corrector aligns the internal state representation with the actual video evolution to prevent model drift, the predictor forecasts the next state based on the corrected representation, and the decoder converts this predicted state back into tokens for further training.

Experiments using the PHYRE dataset (a benchmark for physical reasoning) show that FPTT achieves greater sample efficiency and training stability compared to baseline models like STEVE. The model’s structured approach enables it to generalize well in physical environments simulated with basic Newtonian physics.


In summary, the paper presents a architecture that leverages the strengths of Transformers and slot encoding for efficient and stable world modeling, demonstrating improvements in tasks requiring understanding and predicting object dynamics in a physical environment​

**Strengths:**

- The paper focuses on world modeling which is an important problem.
- The writing and presentation is clean, which makes understanding the paper easy.
- The paper compares efficiency and accuracy, which helps understand the trade-offs.

**Weaknesses:**

- The contribution is not significant having an internal representation of the previous timesteps is common in world modeling architectures, for instance: Dreamer: https://arxiv.org/pdf/2301.04104, Slotformer: https://arxiv.org/pdf/2210.05861. It's unclear to me how this work is a better architecture than Dreamer or Slotformer or other recent works.
- The evaluations and baselines are weak, the paper only compares against STEVE. I dont think STEVE is a fair comparision as their objective was to get interpretable object representations and not necessarily the metric/benchmark paper uses for evaluation. Further the Decoder-Only model seems to perform as well as the proposed architecture on almost all tasks except efficiency.
- Lastly the work only compares on a single benchmark, which is not being used in the baseline works such as STEVE. I think a fair thing to do would be to compare on benchmarks shown in prior baselines, so we assume they are tuned well.

**Questions:**

- How would the architecture compare against Dreamerv3 or Slotformer in world modeling?
- What happens if u try to make the decoder only model more efficient by reducing the number of tokens or dimensionality of the token?
- How would the paper compare against the baselines in benchmarks proposed in Steve or DreamerV3 or SlotFormer?

---

> ### Author Response · Authors · 2024-11-20
>
> Regarding the second question (as the other ones are addressed in the general comment), we kept the decoder-only model with the same number of parameters as the decoder in the complete model for a fair ablation study. Studying the efficiency of the model as the number of parameters varies is left for future work.

---

### Official Review · Reviewer_XiUu · 2024-11-02

**Soundness:** 1
**Presentation:** 1
**Contribution:** 1
**Rating:** 3
**Confidence:** 4

**Summary:**

The paper proposes the Future-Predicting Transformer Triplet (FPTT), an architecture aimed at modeling of physical world dynamics from video data. It employs Transformers to learn object-centric representations, enabling the model to predict physical interactions between objects more effectively. The architecture is tested on synthetic video dataset PHYRE. The authors also perform an ablation study to understand the contribution of different components.

**Strengths:**

The authors address an important problem of physical world modeling using structured latent representations.

**Weaknesses:**

* __Missing slot encodings and object-centricity.__

While the paper includes _slot encoding_ in its title, the approach itself appears to lack this feature. As I understand it, $\Lambda$ was intended to serve as slot encodings, but it is not even referred to as such. From the description, $\Lambda$ seems more like a standard intermediate representation in transformer layers rather than a distinct slot encoding.

In Appendix A2, the authors reference [1] for their transformer implementation, where they also mention using four slots. However, the referenced implementation does not include a parameter for the number of slots, leaving it unclear how slot encodings are actually integrated into the proposed approach, or if they are implemented at all.

Furthermore, the authors state that _"the representation remains opaque and lacks interpretability,"_ which raises questions about the motivation for using _slot encodings_ in the first place.


* __Experimental methodology.__

Firstly, the authors evaluate their model on a single, very simplistic dataset, while using a relatively large number of parameters. This limited evaluation setup may not provide sufficient empirical evidence to support their claims.

A more significant issue lies in their positioning among related works and choice of baselines. The authors overlook most recent related work (e.g., [2, 3, 4, 5]) and rely solely on STEVE as a baseline, aside from variations of their own approach.


* __Presentation.__

In addition to unclear explanations of their approach and its novelty, the authors fail to position it effectively within the existing literature, lacking a comparative analysis with prior work.

All the figures also present issues: they are unnecessarily large, some are in low resolution, and it is often unclear what the authors aim to demonstrate.

\
References:


[1]: Andrej Karpathy. nanoGPT: The simplest, fastest repository for training/finetuning mediumsized GPTs (Generative Pretrained Transformers), 2023. URL https://github.com/karpathy/nanoGPT. \
[2]: Nakano, A., Suzuki, M. and Matsuo, Y., 2023. Interaction-based disentanglement of entities for object-centric world models. In The Eleventh International Conference on Learning Representations.\
[3]: Villar-Corrales, A., Wahdan, I. and Behnke, S., 2023, October. Object-centric video prediction via decoupling of object dynamics and interactions. In 2023 IEEE International Conference on Image Processing (ICIP) (pp. 570-574). IEEE.\
[4]: Wu, Z., Dvornik, N., Greff, K., Kipf, T. and Garg, A., 2022. Slotformer: Unsupervised visual dynamics simulation with object-centric models. arXiv preprint arXiv:2210.05861.\
[5]: Daniel, T. and Tamar, A., DDLP: Unsupervised Object-centric Video Prediction with Deep Dynamic Latent Particles. Transactions on Machine Learning Research.

**Questions:**

* __Slot encodings.__ How and where do the authors utilize slot encodings, and what specific benefits do they offer in this context?

* __Experimental design.__ Why do the authors limit their evaluation to a single dataset? Additionally, what is the rationale for selecting STEVE as the sole baseline, excluding other relevant related works?

---

### Official Review · Reviewer_NsJS · 2024-11-03

**Soundness:** 2
**Presentation:** 3
**Contribution:** 2
**Rating:** 3
**Confidence:** 4

**Summary:**

This paper present a slotted recurrent network model which uses transformers as the main backbone for "world modeling". In this context the resulting model is an "object centric" learning model which cross attends into VQ-VAE encoded input frame, updates the current state and then predicts the next state using a transformer. The model is trained for state prediction (with two variants, either next state prediction or current state prediction) and is demonstrated to mildly work better than a single external baseline (STEVE) and one ablation model (decoder only, where there is not explicit state representation, just prediction and decoding). The experiments are run on a physical reasoning task (PHYRE) and the output is a classification readout.

**Strengths:**

Originality:
The model presented is a very mild variation on previously published works - using VQ-VAE encodings is nice (though probably requires a bit more analysis) and the general recurrent setup is appealing.

Quality:
The proposed model variants (pre and default) are interesting and probably a good step towards analyzing the model's behaviour.

Clarity:
The paper is nicely structured and well written.

Significance:
The context of the work is important, but see below for criticism.

**Weaknesses:**

Unfortunately the paper suffers from several weaknesses.

Experimental validation:
The method is only validated on one task and even on that task results are not very convincing. The models perform very closely to one another and the claims for efficient learning with the model are not well supported.

Analysis:
In general I don't mind when results of a model are not competitive with baselines or ablations as long as there is good analysis of why that is the case, and how this can improve our understanding of the model or problem. Here, however, these are absent - there's very little analysis of what the model learns, how it does that and what determines its performance.

Presentation:
The experimental result figures are not to the level I would expect to see in an ICLR paper - raw training curves are fine if they tell a clear story. Here, however, they do not - there is very little signal there to observe. Export quality is also quite low and does not at the level I would expect.

Novelty:
While usually I don't think novelty is a determining factor for a paper, I feel here this is quite lacking and the proposed model is indeed quite close to existing literature (SAVI++, PARTS and more ). These are cited in the paper, so I have no complaints on that side, but given the generally weak results and analysis I think this hurts the paper.

**Questions:**

My main question is the use of "slot attention" - as far as I can understand there is actually no slot attention in this model, am I right? it seems that the corrector just uses cross-attention and not slot attention? (the difference would be the soft-max axis).

---

### Author Response · Authors · 2024-11-20

We would like to thank all the reviewers for their thorough work and insightful comments.

Let us address some common concerns in this general comment:

Slot attention: it is true that our model does not, in fact, use the exact slot attention paradigm. This work was originally motivated by a desire to apply the same concepts to a more transformer-like structure, but many details got lost in translation. We apologize for the confusion.

Experimental evaluation: given the time and compute restrictions we faced before the submission deadline, we decided to focus on testing the generalization capabilities of our model by having it face many different task variations of the same dataset. We still have our eyes on the Physion dataset and other relevant work, though.

---

### Meta-Review · Area_Chair_m79u · 2024-12-20

**Metareview:**

This paper introduces a framework for world modelling using tokenised (object-centric) representations. A key part of the method is the combination of a transformer architecture with a slot-attention-like mechanism. Experiments are run on the PHYRE simulated physical reasoning benchmark.

The reviewers acknowledged that this paper addresses an important problem, namely whether structured representations can benefit physical world models.

All reviewers raised significant concerns about clarity and quality of presentation, validity and strength of experimental evaluation, and novelty of the method, resulting in a clear reject decision.

**Additional Comments On Reviewer Discussion:**

Reviewer consensus was to reject the paper.

---

### Decision · Program_Chairs · 2025-01-22

Reject